# Genetic Dissection of Spike Productivity Traits in the Siberian Collection of Spring Barley

**DOI:** 10.3390/biom13060909

**Published:** 2023-05-30

**Authors:** Irina V. Rozanova, Yuriy N. Grigoriev, Vadim M. Efimov, Alexander V. Igoshin, Elena K. Khlestkina

**Affiliations:** 1N.I. Vavilov All-Russian Research Institute of Plant Genetic Resources (VIR), 190000 St. Petersburg, Russia; 2Institute of Cytology and Genetics, Siberian Branch of the Russian Academy of Sciences, Lavrentjeva Ave. 10, 630090 Novosibirsk, Russia; 3Siberian Research Institute of Plant Cultivation and Breeding—Branch of Institute of Cytology and Genetics, Siberian Branch of Russian Academy of Sciences, Krasnoobsk, 630501 Novosibirsk, Russia

**Keywords:** barley, row, GWAS, SNP, PLS, meta-analysis, KASP

## Abstract

Barley (*Hordeum vulgare* L.) is one of the most commonly cultivated cereals worldwide. Its local varieties can represent a valuable source of unique genetic variants useful for crop improvement. The aim of this study was to reveal loci contributing to spike productivity traits in Siberian spring barley and to develop diagnostic DNA markers for marker-assisted breeding programs. For this purpose we conducted a genome-wide association study using a panel of 94 barley varieties. In total, 64 SNPs significantly associated with productivity traits were revealed. Twenty-three SNP markers were validated by genotyping in an independent sample set using competitive allele-specific PCR (KASP). Finally, fourteen markers associated with spike productivity traits on chromosomes 2H, 4H and 5H can be suggested for use in breeding programs.

## 1. Introduction

Barley (*Hordeum vulgare* L.) is one of the most commonly cultivated cereals worldwide [1]. As of 2021, the world’s barley area is 51 million hectares, thus ranking fifth after wheat, corn, rice and soybean (http://www.fao.org/faostat/ru/ accessed on 20 March 2022). The total area occupied by this crop in Russia amounts to 8.8 million hectares. The main goal in barley breeding is to combine high yield with better grain quality.

In Russia, areas are mainly occupied by two-row barley: in 2022, 267 spring barley cultivars were present in the state register of breeding achievements approved for research, of which 27 are six-row forms (https://gossortrf.ru/gosreestr/ accessed on 15 February 2023). Six-row barley harbors many traits useful for breeding practices, such as strong culm, resistance to lodging and disease resistance (for example, six-row barley has dominant resistance to net blotch [2]). For further breeding efforts, the introgression of genes controlling these traits into two-row varieties could represent a promising task.

Among the traits that can affect productivity are ear length (EL), grain number (GN), thousand grain weight (TGW) and grain weight per ear (GW). These traits are economically important as they are primary determinants of barley yield. Genes associated with these productivity traits have been found on most barley chromosomes [3], for example: *HvDep1* [4], *Nud* [5] and *VRS* family [6,7,8]. The two-row trait, which is determined by the *VRS* genes, is dominant in barley.

At the end of the twentieth century, the approaches based on the use of diagnostic DNA markers (marker-assisted selection) became an important part of breeding programs. Among DNA markers, SNPs have a high frequency of occurrence in the genome and a relatively high level of polymorphism. NGS-based methods (such as genotyping-by-sequencing (GBS)) allow identifying new SNPs useful for further breeding programs. However, high-density DNA arrays sufficient for detection of novel loci, closely linked to known SNP loci, remain convenient tools for breeders. In barley, such a high-density DNA array is the commercial Illumina 50K barley chip developed in 2017. It allows detection of about 6000 SNPs from the previous 9K chip [9] and new SNPs identified for barley based on exome capture [10].

Due to NGS-based and microarray-based high-throughput genotyping techniques, genome-wide association studies (GWASs) have become the most suitable and widely used approach for investigation of quantitative trait loci in plants. They represent a powerful tool for identifying genomic regions associated with agronomic traits. For successful identification of loci by a GWAS, the sample under study should have a sufficient level of genetic diversity and wide enough range of phenotypic values for studied traits (morphological characteristics, ripeness groups, etc.). As previously mentioned, two barley types are cultivated: two-row and six-row, in both spring and winter. A review of the existing literature showed that the GWAS method is used to study samples consisting of two-row [11,12], six-row [13], or both [8] accessions.

The aim of this study was to reveal loci contributing to spike productivity traits in Siberian spring barley and to develop diagnostic DNA markers for marker-assisted breeding programs. For this purpose, we conducted a genome-wide association study using a panel of 94 barley accessions with follow-up validation of associated markers in an independent sample.

## 2. Materials and Methods

For candidate marker identification, the following analyses were carried out: phenotyping (structural), genotyping and association testing. First, we conducted a GWAS to reveal significant SNPs associated with agronomic traits for three consecutive years (2016–2018). Additionally, we combined the results of association testing over three years using meta-analysis (modified Fisher’s method). After that, we validated the revealed significant markers using independent samples.

### 2.1. Plant Material

The sample was composed taking into account the genetic and phenotypic diversity. In total, 94 accessions from the Siberian spring barley core collection were used. Initially, 68 varieties of two-row barley were selected, which were supplemented with 26 varieties of six-row barley. The selected accessions included 43 cultivars and lines originating from breeding centers of the Siberian Federal District, 31 cultivars from other breeding centers in Russia and 20 cultivars of foreign origin used in breeding programs in Siberia (Appendix A).

Validation was performed on two independent samples: (i) the first sample contained 25 hybrids of the F4 generation from crossing the six-row cultivar Vakula (VIR kat-30983) with the two-row Talan (VIR kat-/ICG kat-8534), the hybrids were chosen from a seed pool; (i) the second included 11 varieties from the VIR collection (VIR kat-2893, kat-7775, kat-15600, kat-26895, kat-30643, kat-30681, kat-4289, kat-9100, kat-17586, kat-19776, kat-30633), 39 selections from the breeding nursery of the first year (generation F4-F5) and 7 varieties of parental forms of hybrid combinations of these selections: Raushan (kat-30592), Preriya (kat-29438), Tanaj (kat-31604), Omskij 88 (kat-30120), G-21435 (selection line of Siberian Research Institute of Plant Cultivation and Breeding), Vorsinskij 2 (kat-31109), Nutans 642 (kat-29891).

### 2.2. Marker Identification

#### 2.2.1. Field Experiment and Phenotyping

The experimental design for each trial was completed with three replications from 2016–2018 years in Novosibirsk, Russia (latitude: 54.853094|longitude: 83.138094). Ninety-four barley accessions were planted in a single replicate. Every replicate consisted of one row 1 m long with a spacing of 30 cm between rows.

To analyze the spike productivity traits and morphobiological characteristics of the accessions, 20 samples were randomly selected from the collected plants of each spring barley (*Hordeum vulgare*) variety. The threshing and counting of grains of selected plants were performed manually. The mass was recorded with an accuracy of tenths of a gram on electronic scales.

Data were recorded for agro-morphological and yield components, including EL, GN, GW and TGW. To assess the distribution of the trait, all the obtained values were divided into three groups: the group with the minimum values included those which did not exceed the difference between the mean trait value and the standard deviation. The group with the maximum value included those that exceeded the sum of the mean value and standard deviation. The remaining values were assigned to the middle group.

#### 2.2.2. Genotyping

The genotyping of Siberian collection samples was carried out by the company TraitGenetics using a Barley 50 K Illumina Infinium iSELECT chip (Gatersleben, Germany). Additional information about 50K barley chip loci was taken from BARLEYMAP (http://floresta.eead.csic.es/barleymap accessed on 20 March 2022) [14]. The markers were assigned to their physical position on the current version with the Morex V3 genome [15] and genetic position on the POPSEQ_2017 map [16]. As the microarray used was developed using Morex V2 genome assembly, some markers were not identified on the V3 version. So, their position was taken into account according to the Morex V2 version [17].

The obtained SNP data were further filtered for (a) a minor allele frequency of 0.1, (b) rate of missing values no more than 10%. As a result, a set of 27 319 markers was selected for the next analysis [18]. Therefore, on average, the marker density was one SNP per 186 kbp (given barley’s size of 5.1 Gb).

#### 2.2.3. Statistical Analysis

The statistical analysis (except GWAS) for all traits was conducted using PAST 2.17 [19]. Multivariate analysis was performed on the measured quantitative traits by using partial least squares (PLS) [20] implemented in PAST 2.17.

The genome-wide association analysis was carried out using the EMMAX software, which implements a mixed linear model that accounts for a genetic relationship of varieties and sample structure. Based on the *p*-values from EMMAX analysis, QQ plots for all features (Appendix A) were built using the R library “qqman”.

To detect significant SNPs we employed a Bonferroni threshold of 3.811 × 10^−6^, based on the significance level (0.05) divided by the effective number of independent tests (13 120) [21], which was calculated using the LD pruning function (--indep-pairwise 100 5 0.999) implemented in PLINK v.1.9 software [22]. Thus, we used −log10 (*p*-value)  =  5.42 as the significance threshold for different traits in our study. In addition, we set a suggestive threshold of *p* < 10^−4^ in order to highlight candidates not reaching the significance level yet still useful for validation purposes.

It is known that the techniques of combining *p*-values from several similar studies (meta-analysis) often help improve statistical power [23]. Therefore, we also combined the *p*-values of three years for each of ten productive characters. As the results are not independent (resulting from the same genotypes each year), we used modified Fisher’s method accounting for the non-independence implemented in the R package “poolr” [24]. It exploits the effective rather than nominal number of tests, thus avoiding overestimation of statistical significance. The effective number of tests was calculated using the Nyholt method [25].

### 2.3. Validation

When significant markers were revealed using the GWAS, candidate SNPs were selected as the most promising. They were validated using KASP analysis on independent samples. For this purpose, conversion to KASP, commercial genotyping and calculation were carried out. Samples of hybrid lines were genotyped using KASP analysis by LGC Genomics (Hoddesdon, UK).

#### 2.3.1. Converting to KASP Markers

After the candidate SNPs were detected, their reference sequence was obtained from the http://plants.ensembl.org/index.html database (accessed on 14 October 2019). Further, sequences were analyzed with the UGENE software. Polymorphic DNA alleles with flanking 101 bp sequences were presented using the format in which 2 allele states of investigated SNPs were divided by the “/” symbol and enclosed in square brackets. The known polymorphic base pairs were identified using standard nomenclature. The 12 markers, which were converted into the KASP markers, were chosen based on the obtained data. KASP genotyping was conducted by LGC Genomics (UK).

#### 2.3.2. Phenotyping Independent Samples and Testing the Diagnostic Value

Phenotyping data were obtained in 2019 from the same experimental location as the main sample. Structural analysis was carried out as described above.

Values for TGW were measured for lines in an independent sample. Two groups were formed: with the minimum (6 lines) and maximum (10 lines) value of the trait.

GW was measured on an independent sample and two groups were also formed: with the minimum and maximum value.

The trait GN in a sample of 25 hybrids was calculated in two ways: (I) varieties were ranked by the number of grains, regardless of the row type. Two groups of varieties were formed: with a small number of grains and with a large number of grains. (II) The varieties were distributed so that in one group with a small number of grains they showed a low value for two-row and for six-row varieties, and in a group with a large number of grains they showed a high value for two-row and for six-row varieties.

For the second sample, the trait GN was calculated using method I.

The EL was the trait (as well as GN) according to which the selection of plants was carried out in an independent sample and extreme values were selected.

Statistical analysis was performed using the Spearman correlation coefficient and the STATISTICA program was used for calculations. Markers in a research locus were tested for their diagnostic value on a set of 25 and 57 genotypes. The *Diagnostic value* (%) was calculated using the following equation:$$Diagnostic\;value=\frac{number\;of\;lines\;showing\;correct\;test\;results}{total\;number\;of\;analyzed\;lines}*100\left(\%\right)$$

Functional enrichment analysis was performed by means of the DAVID resource (https://david.ncifcrf.gov accessed on 14 March 2023) using all genes found within 1 MB of SNPs associated with a trait against a background of all barley genes. The term was taken into account using a multiple testing threshold of 0.05 (q < 0.05).

## 3. Results

### 3.1. Phenotyping 1. Siberian Collection

In this study, 94 accessions were evaluated (location: latitude: 54.853094 | longitude: 83.138094) over 3 years in field trials. The phenotypic stats (minimum, maximum, mean) of four traits are presented in Figure 1 and Table 1.

The structure analysis of the Siberian collection (Table 1) showed the heterogeneity of the sample. The range of traits’ values over the three years reflects the influence of the environment on the studied phenotypes, for example, the maximum value of the SL trait over three years varies from 10.7 to 12.2. However, having approximately the same value range for three years makes it possible to assess the contribution of the genetic component to the development of the trait.

### 3.2. Statistical Analysis

The PLS analysis clearly classified barley genotypes into two main groups (Figure 2).

The two-row and the six-row barley accessions were separated into two groups. As we can see, the traits studied clearly determine the groups.

Association analysis allowed us to reveal sixty-four significant SNPs associated with three of four analyzed productivity traits. Four genomic regions comprising forty-four SNPs were associated with the GN on chromosomes 2H, 3H, 4H and 5H. A total of six SNPs on chromosome 2H associated with the SL were obtained using a meta-analysis and sixteen SNPs were identified on chromosomes 1H, 2H, 4H and 5H associated with the TGW (Table 2).

As a result, several loci associated with TGW were revealed (Table 2): one locus on chromosome 1H (3.19 cM), one locus on chromosome 5H (98.05 cM), two loci on chromosome 4H (one locus was identified in the interval 3.47–4.19 cM using meta-analysis and the second locus at position 23.6 Mbp (interval not determined)) and one locus determined by meta-analysis on chromosome 2H (8.19 cM).

Loci associated with EL were identified only by meta-analysis. In total, three loci were identified on chromosome 2H (0 cM, 56.09–59.49 cM, 67.49 cM).

In total, three loci associated with GN were identified on chromosome 2H (72.59–76.91 cM, 80.03–81.52 cM and 89.8 cM), one locus on chromosome 4H (29.49–31.14 cM) and one locus on chromosome 5H (151.67 cM).

### 3.3. Development and Validation of Diagnostic Markers

Twenty-three SNPs were selected based on GWAS research, on chromosomes 2H (8.19 cM, 59 cM, 67.5 cM, 76.06–76.91 cM, 80.03–81.5 cM 89.8 cM), 4H (3.47–4.19 cM) and 5H (98.5 cM) according to the TGW, EL and GN traits. These SNPs were converted to KASP markers and analyzed using KASP genotyping technology (Appendix A).

### 3.4. Phenotyping 2. Independent Sampling

Next, study lines with extreme values were selected for analysis. The structure analysis data for both independent samples are shown in Table 3.

According to the structure analysis performed on independent samples, it can be seen that the samples are heterogeneous. For example, the value of the EL feature varies from 4 to 13 cm, with an average value of 8.1–8.3 cm in both samples. The value of the TGW is from 36.2 to 72.9 with an average value of 53.0–55.4 g.

### 3.5. KASP Genotyping Results

KASP analysis of 23 markers was carried out on 2 independent samples: the first sample consisted of 25 F4 hybrids and the second sample consisted of 17 cultivars, 1 line and 39 selections from the first-year breeding nursery based on LGC Genomics (Hoddesdon, UK).

The results that were obtained after genotyping of the first sample (performed on hybrids from crossing 2 × 6) with KASP markers show that only 12 of 23 markers were polymorphic for this sample. For ten of them, one allele was fully associated with the two-row type, and the alternative allele was associated with the six-row type. In order to determine which marker would be optimal for further breeding programs, we considered indicators such as Spearman correlation coefficient, the frequency of occurrence (%) of the allele among varieties of an independent sample with a large trait value, the frequency of occurrence of an alternative allele among samples of an independent sample with a small trait value and diagnostic efficiency.

Eight markers (SCRI_RS_166540, JHI-Hv50k-2016-106356, JHI-Hv50k-2016-106731, JHI-Hv50k-2016-106745, JHI-Hv50k-2016-106749, SCRI_RS_4930, JHI-Hv50k-2016-106776, JHI-Hv50k-2016-107364) correlated with the GN trait (R from 0.527 to 0.809, *p* < 0.05), GW (R from 0.464 to 0.793, *p* < 0.05) and TGW trait (R from 0.547 to 0.675, *p* < 0.05). Diagnostic efficiency varied from 88 to 96%, with the Spearman correlation coefficient exceeding 0.603 at *p* < 0.01 for seven markers, among which the marker JHI-Hv50k-2016-107364 had 0.697 (*p* = 0.000076). No significant correlation was found for EL (R from 0.143 to 0.247, *p* > 0.05).

The results obtained after genotyping of the second independent sample with KASP markers, the sample consisting of 18 accessions (of which 4 cultivars were 6-row cultivars) and 39 selections from the breeding nursery of the first year (generation F4–F5), are presented in Table 4.

The calculation of the Spearman correlation coefficient (R = 1, *p* < 0.05) showed a significant correlation between DNA markers and rowing.

Regarding the GN trait, twelve of twenty-three markers (JHI-Hv50k-2016-63482, SCRI_RS_166540, JHI-Hv50k-2016-106356, JHI-Hv50k-2016-106731, JHI-Hv50k-2016-106745) were associated with the trait at the nominal (*p* < 0.05) significance level. The Spearman correlation coefficient varied from 0.32 to 0.51. When ranking into two groups (according to method (I)), nine markers were identified (JHI-Hv50k-2016-98990, SCRI_RS_166540, JHI-Hv50k-2016-106356, JHI-Hv50k-2016-106731, JHI-Hv50k-2016-106745, JHI-Hv50k-2016-106749, SCRI_RS_4930, JHI-Hv50k-2016-106776, JHI-Hv50k-2016-107364). Eight markers, except marker JHI-Hv50k-2016-98990, confirmed the results obtained in the first sample.

Five markers were confirmed for the trait EL in an independent sample (JHI-Hv50k-2016-102654, JHI-Hv50k-2016-106330, JHI-Hv50k-2016-106749, SCRI_RS_4930, JHI-Hv50k-2016-323591) on chromosomes 2H and 5H. The diagnostic efficiency for them ranged from 72% to 100%. However, for markers JHI-Hv50k-2016-102654 and JHI-Hv50k-2016-106330, the Spearman correlation coefficient did not confirm the association (Spearman R = 0.22, *p* = 0.103639, *p* > 0.05 and R = 0.169, *p* = 0.23, *p* > 0.05). For markers JHI-Hv50k-2016-106749, SCRI_RS_4930 and JHI-Hv50k-2016-323591, the Spearman correlation coefficient ranged from 0.406 to 0.453 at *p* < 0.01.

Seven markers were associated with the trait TGW. Spearman correlation coefficient ranged from 0.32 to 0.45 (*p* < 0.05) (SCRI_RS_166540, JHI-Hv50k-2016-106749, SCRI_RS_4930, JHI-Hv50k-2016-107364, JHI-Hv50k-2016-108359, JHI-Hv6-3.349 -Hv50k-2016-323591). Diagnostic efficiency was from 65% to 70%.

Six markers (SCRI_RS_166540, JHI-Hv50k-2016-106356, JHI-Hv50k-2016-106731, JHI-Hv50k-2016-106745, JHI-Hv50k-2016-106749, SCRI_RS_4930) in the interval 76.06-76.91 cM and one marker (JHI-Hv50k-2016-110190) in the interval 89.8 cM were common for the traits GN and TGW.

Two markers (SCRI_RS_4930, JHI-Hv50k-2016-106749) located in the interval 76.91 cM on chromosome 2H were common for all three traits.

Only one marker (JHI-Hv50k-2016-227209) on chromosome 4H (Spearman R = 0.386, *p* = 0.0056, *p* < 0.05) proved to be diagnostic for the trait GW, with diagnostic efficiency of 81%.

According to enrichment analysis, the loci validated in an independent sample are involved in a two-component regulatory system (for the trait GN and for the trait GW). When considering the trait GN, from 125 genes that were found within 1Mb of SNPs, associated with it, 7 significant genes (q < 0.05) are involved in the 2-component regulatory system and 6 (q < 0.05) are involved in posttranslational modification by the attachment of either a single phosphate group or of a complex molecule, such as 5’-phospho-DNA, through a phosphate group.

For the trait GW, of 35 genes 7 were involved in a 2-component regulatory system and 5 participated in posttranslational modification. All genes were located on chromosome 4H (Table 5).

Genes associated with the TGW trait were involved in encoding proteins containing amino acid regions represented by multiple copies. These genes group on chromosomes 2H and 5H (Table 6).

## 4. Discussion

In the current study we analyzed a sample of barley accessions adapted to local conditions of Russia. The gene pool of Russian spring barley cultivars has not been previously explored in such association studies. Therefore, it may represent a promising and valuable source of new productivity loci.

The development of NGS technologies allowed the acquisition of new data and made it possible to implement projects on the sequencing of large and complex plant genomes. The International Barley Genome Sequencing Consortium produced a reference map of the barley genome in 2012, the Morex V2 version [17] based on short reads [40], which was the reference sequence until 2021. This achievement was useful for a wide range of researchers in barley genetics and breeding, particularly in the annotation of new genes and the creation of transgenic lines [41,42]. Next, Morex V3 was created, which is an improvement of the previous version through long-read technology. In the V3 version long contigs contain no gaps in the sequences, giving a nearly complete view of the intergenic space and allowing for in-depth studies [15].

However, a single reference assembly does not reflect intraspecific variability. Currently, there is a first-generation barley pangenome where genotypes of 20 varieties of barley have been examined and which makes previously hidden genetic variation available for genetic research and breeding [43].

The plant breeding technologies developed by the leaders in this field are based on DNA marker-assisted (MAS) approaches. The stage preceding MAS involves the search for DNA marker loci in the crop genome, associated with economically important traits. Changes in a limited number of genes can often lead to substantial phenotypic alterations.

Induced recessive alleles at five major row-type loci can independently convert spikes from two-rowed to six-rowed: *SIX-ROWED SPIKE 1* (*VRS1*), *VRS2*, *VRS3* (syn. *INTERMEDIUM-A*), *VRS4* (syn. *INTERMEDIUM-E*) and *VRS5* (syn. *INTERMEDIUM-C*). Six-rowed barley originated with multiple natural recessive *VRS1* alleles [44] which are now accompanied by a natural allele of *VRS5* (*INT-C.A*) that confers improved lateral grain fill in six-row cultivars. The three other major recessive row-type alleles are not prevalent in current six-row cultivars [45,46,47].

It was shown that all *VRS* genes suppress fertility when carpels and awns appear in developing lateral spikelets [48]. *VRS4*, *VRS3* and *VRS5* act through *VRS1* to suppress fertility, probably by inducing *VRS1* expression. The combined action of *VRS3*, *VRS4* or *VRS5* alleles increased the fertility of lateral spikelets despite the presence of a functional *VRS1* allele. The VRS3 allele caused a loss of spikelet identity and determinacy, improved grain uniformity and increased tillering in the presence of *VRS4*, while *VRS5* led to a decrease in the number of shoots and an increase in grain mass.

The loci obtained in current study using a GWAS were compared with the known locations of the *VRS* genes from the literature (Figure 3).

The *VRS1* gene has physical localization: 652031295–652032562 on chromosome 2H [6]. In total, 25 significant SNPs in the 633.3–655.8 Mbp region were found in our study (72.59–81.52 cM). Thirulogachandar (2017) reported that the *VRS1* gene also controls the leaf area [6].

*VRS2* was previously mapped to chromosome 5HL at 19.0 cM proximal to the *short rachilla hair 1* (*srh1*) locus [47]. The *VRS2* sequences are available from NCBI GenBank under accession codes KX601696 to KX601720. The locus that was found on chromosome 5H in our study (98.5 cM) does not match the preliminary localization of *VRS2*, however, it may be used to detect a minor locus associated with barley productivity.

It was known that locus *VRS3* is located on chromosome 1H. The candidate gene underlying *VRS3* was identified by Wilma van Esse and collegues [49] in the interval 35.7–52.6 cM. The marker revealed in the study (JHI-Hv50k-2016-1755) is located in the 3.19 cM region and it does not match the supposed location of the locus *VRS3*.

The locus *VRS4* is located on chromosome 3H. The exact location of the gene has not been determined. Ludqvist (1991) mapped it on a long arm [50]. Then, Huang and Wu (2011) mapped the locus, which they named PRBS, on the short arm of 3H between SSR markers Bmag0508A and HvLTPPB flanking the genomic interval from 14.3 cM to 24.7 cM [51]. Next, Koppolu (2013) and colleagues reported about *VRS4* on the short arm of chromosome 3H in the interval of 37.17–41.68 cM [46]. The marker JHI-Hv50k-2016-197535, which was found in the current study on chromosome 3H, is located at position 81.02 cM. It does not match any of the candidate loci previously identified in the literature. It is located in the HORVU.MOREX.r3.3HG0297820 gene coding for epidermal patterning factor-like protein (EPFL). Members of the EPFL family play diverse roles in plant growth and development, including the guidance of inflorescence architecture and pedicel length [52].

The *VRS5* allele provides improved cross grain filling in six-row varieties. The *INT-C* gene (*VRS5*) is located on chromosome 4H. In our study a locus was identified on chromosome 4H, and it is located at 17.4–17.9 Mbp (linkage map interval of 29.49–31.14 cM). Ramsay (2011) identified SNPs that showed significant associations with the series type in the 16.7–17.9 Mbp locus (26.19 cM). This region contains several strong candidate genes for *INT-C*, in particular barley orthologs of rice genes, which, respectively, are orthologs of the maize domestication gene *TEOSINTE BRANCHED 1* (*ZmTB1*) [53].

Of the twenty-three markers identified by the GWAS, fourteen SNPs can be used as diagnostics based on the validation results. The markers were grouped according to each trait and an enrichment analysis was carried out for the genes near the SNPs associated with the trait. It was found that the genomic regions associated with the GN, GW and TGW traits are involved in various metabolic processes that affect grain productivity. Thus, genomic regions associated with GN and GW traits are involved in the two-component regulatory system, which is a signaling pathway involved in the perception of mother plant-borne signals by the basal endosperm transfer cells of the developing grain [54] and in the posttranslational modification by the attachment of either a single phosphate group or of a complex molecule, such as 5′-phospho-DNA, through a phosphate group. All identified genes are located on chromosome 4H. The product of the discovered genes is histidine-containing phosphotransfer protein, which belongs to the phosphoproteins. Phosphoproteins are known to participate in the regulation of cellular processes, transport, RNA metabolism, stress response, transcription and translation [55]. Chen et al. (2016) found that phosphorylation of starch granule-binding proteins occurs during all grain developmental stages and plays a critical role in starch biosynthesis [56]. Zhen et al. (2017) identified phosphoprotein which participates in starch biosynthesis [55]. Therefore, it is likely that the genomic regions containing phosphoprotein genes contribute to increased barley grain yield. In our study the SNP JHI-Hv50k-2016-227209 was located in the HORVU.MOREX.r3.4HG0332480 gene, which was identified as a result of gene analysis. Markers JHI-Hv50k-2016-227211 and JHI-Hv50k-2016-227406 are located nearby.

Genomic regions associated with the TGW trait are coding for proteins containing amino acid regions represented by multiple copies. These genes were found on chromosomes 2H and 5H. Out of 18 genes coding for proteins with repeats, 12 participate in immune response (Table 6).

The markers JHI-Hv50k-2016-107364 and JHI-Hv50k-2016-108359 are harbored by the gene HORVU.MOREX.r3.2HG0184560, coding for cysteine-rich repeat secretory protein. This protein family participates in the response to biological stress in plants [57,58]. SNP JHI-Hv50k-2016-110190 was located in the HORVU.MOREX.r3.2HG0188380 gene encoding rop guanine nucleotide exchange factor. This factor serves as a regulator of polar growth and pathogen defense reactions [59]. Markers JHI-Hv50k-2016-352385, JHI-Hv50k-2016-352389, JHI-Hv50k-2016-352390 and JHI-Hv50k-2016-352396 in locus 157.60 cM and marker JHI-Hv50k-2016-323595 in locus 98.05 are located on chromosome 5H in HORVU.MOREX.r3.5HG0526670 and HORVU.MOREX.r3.5HG0501100 genes coding for leucine-rich repeat receptor-like protein kinase family protein. Plant receptor-like kinases (RLKs) are an important class of proteins acting in plant defense responses. RLKs have been identified to be involved in broad-spectrum, elicitor-initiated defense responses and as dominant resistance (R) genes in race-specific pathogen defense. Most defense-related RLKs are of the leucine-rich repeat (LRR) subclass [60]. So, eight markers were also localized in the loci and genes involved in plant immunity.

We also identified the marker JHI-Hv50k-2016-59142 in the HORVU.MOREX.r3.2HG0095740 gene coding for nodulin homeobox, which is one of the negative signaling pathway regulators of the phytohormone abscisic acid [61].

Five markers, JHI-Hv50k-2016-323591, JHI-Hv50k-2016-323593, JHI-Hv50k-2016-323458, JHI-Hv50k-2016-323459 and JHI-Hv50k-2016-323570, were located on chromosome 5H in the locus 98.05 cM. Two of them (JHI-Hv50k-2016-323591, JHI-Hv50k-2016-323593) are in the gene HORVU.MOREX.r3.5HG0501110, coding for putative receptor protein kinase belonging to pattern recognition receptors involved in plant immunity [62]. The markers JHI-Hv50k-2016-323458 and JHI-Hv50k-2016-323459 are located in the gene HORVU.MOREX.r3.5HG0501080 which codes for COS26 protein. The marker JHI-Hv50k-2016-323570 is in the HORVU.MOREX.r3.5HG0501120 gene coding for the 26S proteasome regulatory subunit.

The marker JHI-Hv50k-2016-98990 is located on chromosome 2H at 59.35 cM, according to the genetic map of Mascher et al. (2017) [16]. It is in the HORVU.MOREX.r3.2HG0171930 gene coding for mitochondrial phosphate carrier protein. The phosphate carrier (PiC) is a proton/phosphate symporter which transports negatively charged inorganic phosphate across the inner mt membrane [63]. In the same locus, the marker SCRI_RS_107754 was found based on meta-analysis (chromosome 2H, 59.49 cM).

We determined the location of markers using the Morex V3 map [15]. Three markers, each of which is associated with two of the three traits from Table 4 (SCRI_RS_166540, JHI-Hv50k-2016-106330, JHI-Hv50k-2016-106356), are in the HORVU2Hr1G091030 and HORVU2Hr1G091170 genes. The authors of [64] reported that these genes were associated with grain number.

## 5. Conclusions

Our study represents a search for loci related to barley spike productivity traits using the unique Siberian barley germplasm. We performed an experiment lasting for three consecutive years (2016, 2017, 2018) under different natural conditions (multienvironmental, multiyear) and studied the effect of rowing on such agronomic traits as ear length (EL), grain number (GN), grain weight per ear (GW) and thousand grain weight (TGW). It was found that the row-type genes make a large contribution in the combined samples. Eleven chromosome regions were identified using a GWAS. Four regions were explored in more detail. Twenty-three markers from these four regions were chosen for KASP genotyping. Fourteen of them were confirmed. Only one of these fourteen loci was previously described in the literature. Thus, the GWAS-based approach using previously unexplored material and high-density microarray for genotyping is an effective way for identifying new loci, even with a modest sample size. The results obtained open up the possibility of more efficient use of Siberian germplasm for breeding barley with high spike productivity.

## Figures and Tables

**Figure 1 biomolecules-13-00909-f001:**
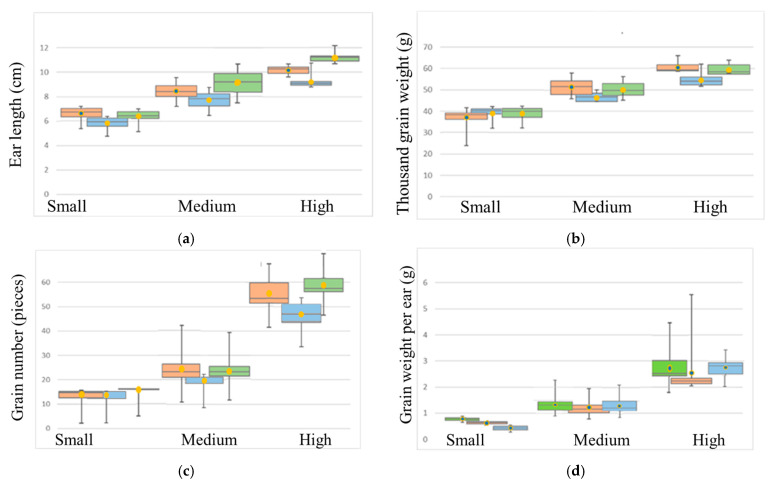
Box plots for distribution of values for the studied phenotypic traits: (**a**) ear length (EL); (**b**) thousand grain weight (TGW); (**c**) grain number (GN); (**d**) grain weight per ear (GW). Data obtained in 2016 are shown in orange, in 2017 in blue and in 2018 in green.

**Figure 2 biomolecules-13-00909-f002:**
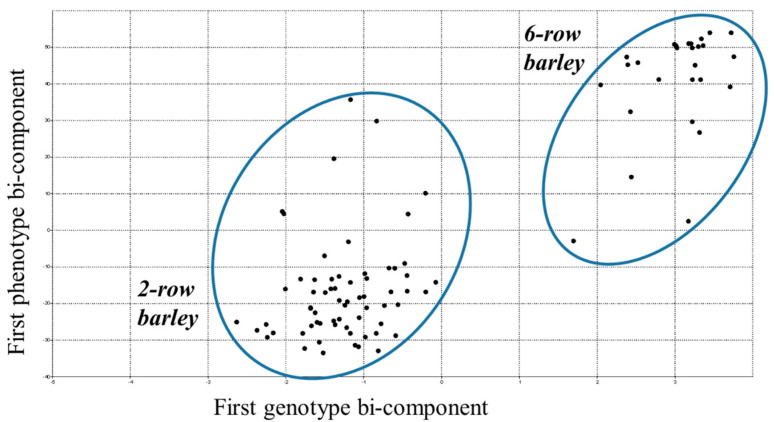
Partial least squares based on genetic distance calculated for 94 accessions and the results of three-year measurements of four agronomic traits. The analysis clearly shows that two clusters correspond to two row types.

**Figure 3 biomolecules-13-00909-f003:**
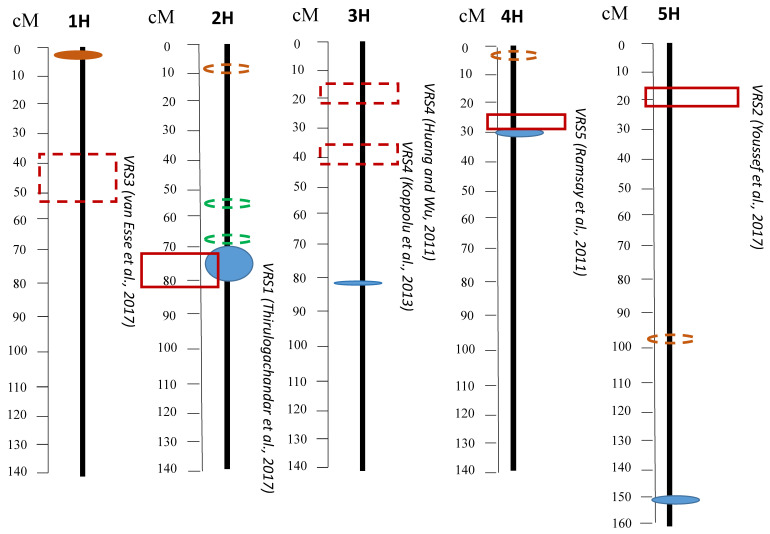
Identified loci and localization of *VRS* genes. Filled ellipses denote loci whose *p*-value passed the threshold value, dotted lines denote loci classified as suggestive. The boxes show the known localization of the *VRS* genes.

**Table 1 biomolecules-13-00909-t001:** Descriptive statistics by year for studied phenotypic traits in Siberian collection.

Trait ^1^	Mean over Three Years			Max			Min		
	2016	2017	2018	2016	2017	2018	2016	2017	2018
EL	7.6 + 1.2	9 + 1.6	8.4 + 1.2	10.7	12.2	10.7	4.7	5.0	5.3
GN	27 + 12	30 + 15	32 + 16	54	68	72	11	12	15
GW	1.4 + 0.6	1.5 + 0.7	1.6 + 0.7	5.5	3.4	3.7	0.5	0.4	0.7
TGW	48.3 + 6.7	49.4 + 8.6	49.6 + 7.2	62.0	65.9	63.9	30.4	21.5	29.4

^1^ EL: ear length, GN: grain number, GW: grain weight per ear, TGW: thousand grain weight.

**Table 2 biomolecules-13-00909-t002:** Summary of association mapping results for 4 agronomic traits.

Trait	Marker	Chr	Position	Interval	*p*-value
GN	BOPA2_12_31445 ^2017,2018^ *	2	560716546	72.59	4.99 × 10^−7^
GN	JHI-Hv50k-2016-105183 ^2016,2018^ *	2	561924540	73.73	3.31 × 10^−6^
GN	JHI-Hv50k-2016-106221 ^2016^ *^,2017,2018^	2	566277088	75.57	9.70 × 10^−7^
GN	JHI-Hv50k-2016-106223 ^2016,2017,2018^	2	566277009	75.57	3.59 × 10^−7^
GN	JHI-Hv50k-2016-106229 ^2016^ *^,2017,2018^	2	566606207	76.06	9.70 × 10^−7^
GN	JHI-Hv50k-2016-106267 ^2016,2017,2018^	2	566875752	76.2	3.65 × 10^−7^
GN	JHI-Hv50k-2016-106268 ^2016,2017,2018^	2	566877197	76.2	9.02 × 10^−9^
GN	SCRI_RS_166540 ^2016,2018^	2	566932140	76.06	4.00 × 10^−7^
GN	JHI-Hv50k-2016-106330 ^2016,2018^	2	567013913	76.06	4.00 × 10^−7^
GN	JHI-Hv50k-2016-106356 ^2016,2018^	2	566985374	76.06	4.00 × 10^−7^
GN	JHI-Hv50k-2016-106374 ^2016^ *^,2017,2018^	2	567080995	76.06	4.29 × 10^−7^
GN	JHI-Hv50k-2016-106377 ^2016,2017^ *^,2018^	2	567079310	76.06	1.64 × 10^−6^
GN	JHI-Hv50k-2016-106459 ^2016^ *^,2017^ *^,2018^	2	567608633	76.91	5.46 × 10^−7^
GN	JHI-Hv50k-2016-106465 ^2016^ *^,2017^ *^,2018^	2	567606437	76.91	5.46 × 10^−7^
GN	JHI-Hv50k-2016-106526 ^2016^ *^,2017^ *^,2018^	2	567666862	76.91	5.46 × 10^−7^
GN	JHI-Hv50k-2016-106554 ^2016^ *^,2017^ *^,2018^	2	567813910	76.77	5.46 × 10^−7^
GN	JHI-Hv50k-2016-106731 ^2016,2017,2018^	2	568177791	76.91	1.35 × 10^−7^
GN	JHI-Hv50k-2016-106745 ^2016,2017,2018^	2	568184982	76.91	1.87 × 10^−6^
GN	JHI-Hv50k-2016-106449 ^2016^ *^,2017,2018^	2	567426651	NA	1.93 × 10^−8^
GN	JHI-Hv50k-2016-106749 ^2016^ *^,2017,2018^	2	568186319	NA	1.93 × 10^−8^
GN	SCRI_RS_4930 ^2016,2017,2018^	2	568413596	76.91	1.35 × 10^−7^
GN	JHI-Hv50k-2016-106776 ^2016,2017,2018^	2	568593779	76.91	1.35 × 10^−7^
GN	JHI-Hv50k-2016-107364 ^2016,2017,2018^	2	570022801	80.03	2.39 × 10^−9^
GN	JHI-Hv50k-2016-108359 ^2016^ *^,2017,2018^	2	573567807	80.1	1.49 × 10^−8^
GN	JHI-Hv50k-2016-108474 ^2016^ *^,2017,2018^	2	573814235	81.52	1.49 × 10^−8^
GN	JHI-Hv50k-2016-110190 ^2016,2017^ *^,2018^	2	591535263	89.8	7.09 × 10^−7^
GN	JHI-Hv50k-2016-197535 ^2017^ *^,2018^	3	531392349	81.02	3.92 × 10^−7^
GN	JHI-Hv50k-2016-230985 ^2016^ *^,2018^	4	15576552	29.49–31.14	2.19 × 10^−6^
GN	JHI-Hv50k-2016-230986 ^2016^ *^,2018^	4	15576904	29.49–31.14	2.19 × 10^−6^
GN	JHI-Hv50k-2016-231001 ^2016^ *^,2018^	4	15592997	29.49–31.14	2.19 × 10^−6^
GN	JHI-Hv50k-2016-231004 ^2016^ *^,2018^	4	15593574	29.49–31.14	2.19 × 10^−6^
GN	JHI-Hv50k-2016-231008 ^2016^ *^,2018^	4	15593889	29.49–31.14	2.19 × 10^−6^
GN	JHI-Hv50k-2016-231027 ^2016^ *^,2018^	4	15657353	29.49–31.14	2.19 × 10^−6^
GN	JHI-Hv50k-2016-231038 ^2016^ *^,2018^	4	15816073	29.49–31.14	2.19 × 10^−6^
GN	JHI-Hv50k-2016-231053^2016^*^,2018^	4	16055421	29.49–31.14	2.19 × 10^−6^
GN	JHI-Hv50k-2016-231059 ^2016^ *^,2018^	4	16054056	29.49–31.14	2.19 × 10^−6^
GN	JHI-Hv50k-2016-231067 ^2016^ *^,2018^	4	16052679	29.49–31.14	2.19 × 10^−6^
GN	JHI-Hv50k-2016-232164 ^2016^ *^,2017^ *^,2018^	4	20926211	29.49–31.14	2.43 × 10^−6^
GN	JHI-Hv50k-2016-232178 ^2016^ *^,2017^ *^,2018^	4	20923563	29.49–31.14	2.43 × 10^−6^
GN	JHI-Hv50k-2016-352385 ^2017^ *^,2018^	5	564103520	151.67	2.54 × 10^−6^
GN	JHI-Hv50k-2016-352389 ^2017^ *^,2018^	5	564103938	151.67	2.54 × 10^−6^
GN	JHI-Hv50k-2016-352390 ^2017^ *^,2018^	5	564103970	151.67	2.54 × 10^−6^
GN	JHI-Hv50k-2016-352396 ^2017^ *^,2018^	5	564104465	151.67	2.54 × 10−6
GN	JHI-Hv50k-2016-352506 ^2017^ *^,2018^	5	564298766	151.53	2.54 × 10^−6^
EL	JHI-Hv50k-2016-59142 ^meta^	2	648002	0	3.30 × 10^−5^ *
EL	JHI-Hv50k-2016-90345 ^meta^	2	155675417	56.09	2.62 × 10^−5^ *
EL	JHI-Hv50k-2016-98990 ^meta^	2	502130218	59.35	2.37 × 10^−5^ *
EL	SCRI_RS_107754 ^meta^	2	NA *	59.49	8.39 × 10^−5^ *
EL	JHI-Hv50k-2016-102654 ^meta^	2	545723106	67.49	5.30 × 10^−5^ *
EL	JHI-Hv50k-2016-102655 ^meta^	2	545723259	67.49	5.30 × 10^−5^ *
TGW	JHI-Hv50k-2016-1755 ^2016^	1	1710185	3.19	1.19 × 10^−6^
TGW	JHI-Hv50k-2016-63480 ^meta^	2	9048890	8.19	6.96 × 10^−5^ *
TGW	JHI-Hv50k-2016-63482 ^meta^	2	9046542	8.19	6.96 × 10^−5^ *
TGW	JHI-Hv50k-2016-227209 ^meta^	4	3317320	4.19	8.95 × 10^−5^ *
TGW	JHI-Hv50k-2016-227211 ^meta^	4	NA *	NA	8.95 × 10^−5^ *
TGW	JHI-Hv50k-2016-227406 ^meta^	4	3348318	3.47	8.95 × 10^−5^ *
TGW	JHI-Hv50k-2016-232164 ^meta^	4	20926211	29.49–31.14	6.35 × 10^−5^ *
TGW	JHI-Hv50k-2016-232178 ^meta^	4	20923563	29.49–31.14	6.35 × 10^−5^ *
TGW	JHI-Hv50k-2016-323458 ^meta^	5	506072932	98.05	3.32 × 10^−5^ *
TGW	JHI-Hv50k-2016-323459 ^meta^	5	506072959	98.05	3.32 × 10^−5^ *
TGW	JHI-Hv50k-2016-323570 ^meta^	5	506120371	98.05	3.32 × 10^−5^ *
TGW	JHI-Hv50k-2016-323591 ^meta^	5	506116732	98.05	3.32 × 10^−5^ *
TGW	JHI-Hv50k-2016-323593 ^meta^	5	506115673	98.05	3.32 × 10^−5^ *
TGW	JHI-Hv50k-2016-323595 ^meta^	5	506115168	98.05	3.32 × 10^−5^ *

The column *p*-value indicates the smallest value obtained; the marker superscripted stand for analyzed years, meta—meta-analysis over three years; * suggestive (values are low enough, but not exceeding the threshold) physical positions were determined with V2 [17].

**Table 3 biomolecules-13-00909-t003:** Descriptive statistics of studied phenotypic traits of independent samples.

Trait ^1^	Independent Sample 1 (F4 from Cross 2 Row × 6 Row)			Independent Sample 2 (Hybrids F4–F5)		
	Mean	Max	Min	Mean	Max	Min
EL	8.3 ± 2.3	12	4	8.1 ± 1.3	13	4.6
GN	37 ± 21	77	9	23 ± 4	47	10
WS	2.0 ± 1.0	4.3	1.8	1.2 ± 0.3	1.8	0.6
TGW	55.4 ± 7.8	72.9	44.9	53.0 ± 7.8	67.8	36.2

^1^ EL: ear length, GN: grain number, GW: grain weight per ear, TGW: thousand grain weight.

**Table 4 biomolecules-13-00909-t004:** The KASP genotyping results and the allelic ratios for markers associated in independent sample of 18 accessions and 39 selections from the breeding nursery of the first generation (F4–F5). Allelic frequencies are indicated for contrast groups (X, Y). The KASP primers for the studied markers are given in the Appendix A.

Marker	Trait	Allele Frequency X	Allele Frequency Y	Chr	Position
JHI-Hv50k-2016-63480 ^meta^	Not identified as diagnostic	0.81	0.59	2	9048890
JHI-Hv50k-2016-63482 ^meta^	Not identified as diagnostic	0.81	0.59	2	9046542
JHI-Hv50k-2016-98990 ^meta^	GN	0.91	0.07	2	502130218
SCRI_RS_107754 ^meta^	Not identified as diagnostic	0.72	0.44	2	NA *
JHI-Hv50k-2016-102654 ^meta^	Not identified as diagnostic	0.63	0.04	2	545723106
JHI-Hv50k-2016-102655 ^meta^	Not identified as diagnostic	0.63	0.04	2	545723259
SCRI_RS_166540 ^2016,2018^	TGW GN	0.91	0.11	2	566932140
JHI-Hv50k-2016-106330 ^2016,2018^	Not identified as diagnostic	0.91	0.11	2	567013913
JHI-Hv50k-2016-106356 ^2016,2018^	GN	0.91	0.11	2	566985374
JHI-Hv50k-2016-106731 ^2016,2017, 2018^	GN	0.88	0	2	568177791
JHI-Hv50k-2016-106745 ^2016,2017,2018^	GN	0.90	0	2	568184982
JHI-Hv50k-2016-106749 ^2016,2017,2018^	EL TGW GN	0.85	0	2	568186319
SCRI_RS_4930 ^2016,2017,2018^	EL TGW GN	0.88	0	2	568413596
JHI-Hv50k-2016-106776 ^2016,2017,2018^	GN	0.88	0	2	568593779
JHI-Hv50k-2016-107364 ^2016,2017,2018^	TGW GN	0.99	0.15	2	570022801
JHI-Hv50k-2016-108359 ^2016,2017,2018^	TGW	0.97	0.11	2	573567807
JHI-Hv50k-2016-108474 ^2016,2017,2018^	Not identified as diagnostic	0.97	0.11	2	573814235
JHI-Hv50k-2016-110190 ^2016,2017,2018^	TGW GN	0.91	0.07	2	591535263
JHI-Hv50k-2016-227209 ^meta^	GW	0.31	0	4	3317320
JHI-Hv50k-2016-227406 ^meta^	Not identified as diagnostic	0.31	0	4	3348318
JHI-Hv50k-2016-323458 ^meta^	Not identified as diagnostic	0.94	0.48	5	506072932
JHI-Hv50k-2016-323459 ^meta^	TGW	0.94	0.48	5	506072959
JHI-Hv50k-2016-323591 ^meta^	EL, TGW	0.94	0.48	5	506116732

The marker superscripted stand for analyzed years, meta—meta-analysis over three years; * physical position was determined with V2 [17].

**Table 5 biomolecules-13-00909-t005:** Genes underlying statistically significant terms enriched according to DAVID analysis for GN and GW traits.

Genes	Product	Term
HORVU.MOREX.r3.4HG0332510 HORVU.MOREX.r3.4HG0332490 HORVU.MOREX.r3.4HG0332480 HORVU.MOREX.r3.4HG0332500 HORVU.MOREX.r3.4HG0332520 HORVU.MOREX.r3.4HG0332600	Histidine-containing phosphotransfer protein 2-like	Two-component regulatory system
HORVU.MOREX.r3.4HG0333000	General negative regulator of transcription subunit 3	Phosphoprotein
HORVU.MOREX.r3.4HG0332600 HORVU.MOREX.r3.4HG0332490 HORVU.MOREX.r3.4HG0332500 HORVU.MOREX.r3.4HG0332520	Histidine-containing phosphotransfer protein 2-like	Phosphoprotein

**Table 6 biomolecules-13-00909-t006:** Genes underlying statistically significant terms enriched according to DAVID analysis for TGW trait.

Genes	Product	Function
HORVU.MOREX.r2.2HG0155940, HORVU.MOREX.r2.2HG0155940, 1HORVU.MOREX.r3.2HG0188290	Aquaporin PIP1-1-like	Immune response [26]
HORVU.MOREX.r3.2HG0183520	MDIS1-interacting receptor-like kinase 2-like	Immune response [27]
HORVU.MOREX.r3.2HG0183710	WD repeat-containing protein 44-like	Immune response [28]
HORVU.MOREX.r3.5HG0501670	Ankyrin repeat-containing protein ITN1-like	Immune response [29]
HORVU.MOREX.r3.2HG0183650	Annexin-like protein RJ4	Immune response [30]
HORVU.MOREX.r3.2HG0188200	Calcium-dependent protein kinase 12-like	Immune response [31]
HORVU.MOREX.r3.5HG0501470, HORVU.MOREX.r3.5HG0501480, HORVU.MOREX.r3.5HG0501530, HORVU.MOREX.r3.2HG0184560	Cysteine-rich receptor-like protein kinase 6	Immune response [32]
HORVU.MOREX.r3.2HG0184300	Probable LRR receptor-like serine/Threonine-protein kinase At1g63430	Immune response [33]
HORVU.MOREX.r2.5HG0415670, HORVU.MOREX.r2.5HG0415670.1, HORVU.MOREX.r3.5HG0500800	Probable aquaporin PIP2-7	Immune response [34]
HORVU.MOREX.r2.2HG0151840, HORVU.MOREX.r3.2HG0183550	Probable aquaporin TIP3-2	Immune response [35]
HORVU.MOREX.r3.5HG0500740	Histone-binding protein MSI1 homolog	Diverse chromatin-associated complexes [36], histone-binding proteins recognize (“read”) certain histone residues and their modifications [37]
HORVU.MOREX.r3.2HG0184620	Hsp70–Hsp90 organizing protein-like	Transport function [38]
HORVU.MOREX.r3.5HG0501330	Pentatricopeptide repeat-containing protein At1g20300, mitochondrial	The function of this gene is completely unknown, but homologs from other plant species are involved in the control of male fertility by regulating expression of mitochondrial “sterility” genes [39]
HORVU.MOREX.r3.2HG0185390	Uncharacterized LOC123427401 (calcium-dependent protein kinase, putative)	Unknown (presumably immune)
HORVU.MOREX.r3.2HG0184310	Uncharacterized protein DDB_G0286299-like	Unknown

## Data Availability

The following information was supplied regarding data availability: Genotyping data and raw data on the mean values are available in the Appendix A.

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
