# Peer review of "Genetic Dissection of Spike Productivity Traits in the Siberian Collection of Spring Barley"

_biomolecules, 2023, doi:10.3390/biom13060909_

Round 1
Reviewer 1 Report
Thanks authors to submit the paper to MDPI. For me, the scope of lants journal would be more appropriate.
Authors attemp to describe potential of siberian barley cultivars/lines. These may be precious genetic resources The materials were characterized by a standard way. Results adequatly described.
I wish the author discuss the analytic tool 50K barley chip, whether more information and new variants could be achievet by NGS.
Discussion could be more vivid and interesting when functionality of the genes is more deep described

OK with respect to grammar
Author Response
Dear Reviewer,
thank you for considering the contribution to Biomolecules entitled " Genetic dissection of spike productivity traits in the Siberian collection of spring barley " by Irina V. Rozanova, Yuriy N. Grigoriev, Vadim M. Efimov, Alexander V. Igoshin and Elena K. Khlestkina.
The paper has been reworked following point-by-point the referee’s comments.

Reviewer 2 Report
As the title expresses, the research was aimed at genomic assessment of spike productivity traits using Siberian collection of barley. Since the access to the specific germ plasm is quite difficult due to ownership issues and IPR, the info is valid for the users of barley as a major crop out of Russia whether the collection could have some unique value with the genomic assessment on the spike traits as yield components.
The research approach is orthodox and not very original, but the materials are unique, and this support the value of the manuscript for barley breeding and genetics community which has a wide audience globally.
The collection from Siberian region is unique and unexploited for the users out of Russia.
The methodology is fine but as authors use multi-year and multi-location testing and, it could add accuracy of what they analyze on genotypes x environment interactions.
The result is rather predicted that there is some specific genomic region(s )responsible for the spike traits that would be coincide with the barley genome initiatives such as:
http://plants.ensembl.org/Hordeum_vulgare/Info/Index
https://www.naro.affrc.go.jp/archive/nias/eng/genome/oomugi/
https://www.nature.com/articles/s41586-020-2947-8
Could be more and small review can be made to add up the major barley initiatives.
Table and figure legends could be more descriptive to make more self-explanatory of each item.
The language is appropriate, but some sentences can be rewritten for more clarity with shorter sentences to avoid ambiguity.
Author Response
Dear Reviewer,
Thank you for your comments!
As the title expresses, the research was aimed at genomic assessment of spike productivity traits using Siberian collection of barley. Since the access to the specific germ plasm is quite difficult due to ownership issues and IPR, the info is valid for the users of barley as a major crop out of Russia whether the collection could have some unique value with the genomic assessment on the spike traits as yield components. The research approach is orthodox and not very original, but the materials are unique, and this support the value of the manuscript for barley breeding and genetics community which has a wide audience globally. The collection from Siberian region is unique and unexploited for the users out of Russia. The methodology is fine but as authors use multi-year and multi-location testing and, it could add accuracy of what they analyze on genotypes x environment interactions.
The result is rather predicted that there is some specific genomic region(s )responsible for the spike traits that would be coincide with the barley genome initiatives such as:
http://plants.ensembl.org/Hordeum_vulgare/Info/Index
https://www.naro.affrc.go.jp/archive/nias/eng/genome/oomugi/
https://www.nature.com/articles/s41586-020-2947-8
Could be more and small review can be made to add up the major barley initiatives.
Response 1. Thank you. We included information about these initiatives in lines 329 – 342. «The NGS-technologies development allowed to obtain new data and made it possible to implement projects on the sequencing of large and complex plant genomes. The International Barley Genome Sequencing Consortium produced a reference map of the barley genome in 2012, the Morex V2 version [17] based on short reads [40] which was the reference sequence until 2021. This achievement was useful for a wide range of researchers in barley genetics and breeding, particularly in the annotation of new genes and the creation of transgenic lines [41, 42]. Next, Morex V3 was created, which is an improvement of the previous version through long-read technology. In the V3 version long contigs contain no gaps in the sequences, giving a nearly complete view of the intergenic space and allowing for in-depth studies [15].
However, a single reference assembly does not reflect intraspecific variability. Currently, there is a first-generation barley pangenome where genotypes of 20 varieties of barley have been examined and which makes previously hidden genetic variation available for genetic research and breeding [43].»
Table and figure legends could be more descriptive to make more self-explanatory of each item.
Response 2. Thank you, we have specified more detail in the revised MS .
Reviewer 3 Report
In this study, the authors found 64 SNPs significantly associated with productivity traits, and 14 KASP markers were identified for use in breeding programs.
1、 It was seen that 94 barley varieties were not enough to dissect the genetic basis of spike productivity traits in spring barley, more barley varieties were suggested to be used for GWAS.
2、 Figure 1 has no units on the axes.
This manuscript contains some mistakes and errors in grammar and format. The manuscript needs substantial improvement in both writing and the use of good English.
Author Response
Dear Reviewer,
Thank you for your comments!
In this study, the authors found 64 SNPs significantly associated with productivity traits, and 14 KASP markers were identified for use in breeding programs.
Point 1. It was seen that 94 barley varieties were not enough to dissect the genetic basis of spike productivity traits in spring barley, more barley varieties were suggested to be used for GWAS.
Response 1. The collection from Siberian region is unique and unexploited for the users out of Russia. Despite the modest sample size, we were aware that we are using a previously unexplored material, a unique germplasm. This provided the basis for the identification of new loci, despite the small sample size.
Point 2. Figure 1 has no units on the axes.
Response 2. We have specified this in the revised MS.
Point 3. Comments on the Quality of English Language
This manuscript contains some mistakes and errors in grammar and format. The manuscript needs substantial improvement in both writing and the use of good English.
Response 3. The language was improved.
Round 2
Reviewer 3 Report
The authors have replied to all the questions and improved the quality of the manuscript, suggesting that it be accepted for publication.
Minor editing of English language required.